# Assessment tools for transition readiness in adolescents with inflammatory bowel disease: A scoping review

YaHui Zuo[1,2], Mei Li[2☯]*, JinJin Cao[3☯]*, JiaNan Wang[2,4], WenQian Cai[2,4], Lu Zhang[1], Meng Li[5]

1 School of Nursing, Nanjing Medical University, Nanjing, Jiangsu Province, China, 2 Department of Nursing, Children's Hospital of Nanjing Medical University, Nanjing, Jiangsu Province, China, 3 Department of Nursing, Nanjing BenQ Medical Center, The Affiliated BenQ Hospital of Nanjing Medical University, Nanjing, Jiangsu Province, China, 4 School of Pediatrics, Nanjing Medical University, Nanjing, Jiangsu Province, China, 5 Department of Rehabilitation Medicine, Children's Hospital of Nanjing Medical University, Nanjing, Jiangsu Province, China

☯ These authors contributed equally to this work.
* limei_njch@163.com (ML); 776084094@qq.com (JC)

**Data Availability Statement:** All relevant data are within the manuscript and its Supporting information files.

## Abstract

### Background

Assessing the level of transition readiness in adolescents with inflammatory bowel disease is crucial; however, standardized research tools are lacking. This study aimed to map transition readiness assessment tools for adolescents with inflammatory bowel disease and determine their suitability.

### Methods

A literature review following the Arksey and O'Malley scoping review methodology was conducted. By using appropriate key terms, literature on transition readiness assessment tool searches were conducted in the CNKI, WanFang, SinoMed, Pubmed, Cochrane Library, Web of Science, and CINAHL databases, with a reference search. The retrieval period was from the establishment of the databases to January 2024.

### Results

A total of 2561 studies were obtained through a preliminary search, and 5 references were obtained as retrospective references. Finally, 21 studies were selected for this review. In total, 20 transition readiness assessment tools were identified. Qualitative findings were grouped into five thematic areas: descriptive characteristics of reviewed articles, development procedures, design, psychometric properties, and cohort characteristics for validity testing of transition readiness assessment tools.

### Conclusions

The most appropriate way to assess the transition readiness of adolescents with inflammatory bowel disease is to select an assessment tool that is most suitable for individual needs,

**Funding:** The author(s) received no specific funding for this work.

**Competing interests:** The authors have declared that no competing interests exist.

accompanied by a comprehensive patient evaluation. Despite some flaws in the methodology, TRM is currently the most suitable assessment tool, and more population studies are needed to validate it.

## Introduction

Inflammatory bowel disease (IBD) is a lifelong, nonspecific chronic gastrointestinal inflammatory disease. It comprises three primary subtypes: Crohn's disease (CD), ulcerative colitis (UC), and IBD-unclassified (IBD-U) [1, 2]. The disease course of IBD is characterized by remitting and relapsing symptoms, which vary significantly between individuals. Notably, one-quarter of patients with IBD are diagnosed during childhood, and the incidence of this disease among children is on the rise [3]. With an increase in the number of children diagnosed with IBD, there is a growing number of young patients who must transition to the adult healthcare system. Transition Readiness is the ability of youth and their support system to transition from pediatric to adult health care system successfully [4]. It is usually used as an indicator of the healthcare transition process for children, reflecting the level of self-management ability, which has important predictive significance for quality of life with respect to disease after the transition [5]. Gumidyala [6] discovered that the majority of adolescents with IBD were not adequately prepared for transition, resulting in lower chances of successful transition, increased rates of emergency room visits, hospitalizations, and surgeries, and reduced quality of life. This significantly impacted their education, employment, and social integration. Bhawra [7] found that implementing effective transitional care can reduce emergency room admissions for adolescents with chronic conditions, save healthcare costs, and improve the health-related quality of life. Therefore, it is crucial to ensure a smooth transition of care for these patients as they enter adulthood with this complex illness. The first step in implementing transitional care is to conduct a comprehensive assessment of transition readiness [8]. Medical service providers should choose a suitable assessment tool to evaluate the level of transition preparation for children with IBD. It will help guide clinical decision-making and enable targeted nursing measures to improve the transition readiness. Hence, we can improve the quality and efficiency of medical services and reduce the wastage of medical resources [9]. To date, there are several tools available for assessing transition readiness, they are widely utilized in adolescents with a variety of chronic diseases, including digestive disorders [10, 11]. However, these transition readiness assessment tools have some shortcomings. For example, the Transition Readiness Assessment Questionnaire(TRAQ) [12] is limited to skill aspects. There is a lack of research that systematically reviews transition readiness assessment tools for adolescents with IBD. This has resulted in difficulties in the selection of appropriate tools. Therefore, the aim of this study was to map transition readiness assessment tools for children with IBD and identify deficiencies in the psychometric properties, applicability, and reliability of assessment tools.

## Methods

### Protocol

The review protocol followed the Arksey and O'Malley method of scoping review and JBI scoping review guidance [13]. The stages are as follows: research question, identifying relevant studies, study selection, charting the data, collating, summarizing, and reporting the results.

### Research question

The following research questions were identified through a previous literature review: ① What are the current transition readiness assessment tools for IBD patients? ② How are the reliability and validity of each transition readiness assessment tool applied to IBD patients? ③ How is the IBD transition readiness assessment tool applied?

### Identifying relevant studies

**Information sources.** The following electronic databases were searched: CNKI, Wan-Fang, SinoMed, Pubmed, Cochrane Library, Web of Science, and CINAHL. The last search date was 1st January 2024.

**Search strategy.** The combination of Medical Subject Headings (MeSH) terms and free words was used to search the 7 abovementioned Chinese and English databases. The keywords searched were (("Health Transition") OR ("Transition to Adult Care") OR ("Transitional Care") OR ("transition readiness"))AND("adolescen*" OR "children" OR ("young adult*")) AND ("access" OR "measure" OR "questionnaire" OR "tool" OR "scale" OR "list"). The research team conducted a presearch in PubMed and CNKI and then analyzed and discussed the search results and adjusted the retrieval strategy for formal retrieval.

### Study selection

**Inclusion and exclusion criteria.** The inclusion criteria were as follows: ① the study population included pediatric children with IBD; ② the study involved the development, validation, revision, translation or cross-cultural adaptation of the transition readiness assessment tools; ③ the study type was a quantitative or qualitative study; and ④ the study language was Chinese or English.

The exclusion criteria were as follows: ① the full text could not be obtained; ② the abstract of a meeting; and ③ the collection of literature was repeated.

**Screening process.** The retrieved literature titles were imported into Zotero software to screen repeated documents. According to the inclusion and exclusion criteria, the titles and abstracts were screened by two researchers alone. Finally, the articles that met the inclusion criteria were imported into full-text attachments for full-text reading.

### Charting the data

Researchers independently and by pairs independently extracted the data and information and checked it. Disagreements were resolved by a third reviewer. The following data were extracted: developer, publication date, country/region, scoring method, demarcation value, number of dimensions, number of items, reliability and validity, and tool characteristics.

## Results

### Collating, summarizing, and reporting the results

A total of 2561 studies were obtained through a preliminary search, and 5 references were obtained as retrospective references. Two independent evaluators screened the studies and obtained the same results based on preestablished inclusion and exclusion criteria, leading to the inclusion of 21 articles [12, 14–33]. Fig 1 shows the screening process according to the Preferred Reporting Items for Systematic Reviews and Meta-Analyses (PRISMA) model. The outcomes were grouped into five thematic areas: descriptive characteristics of reviewed articles, development procedures, design, psychometric properties, and cohort characteristics for validity testing of transition readiness assessment tools.

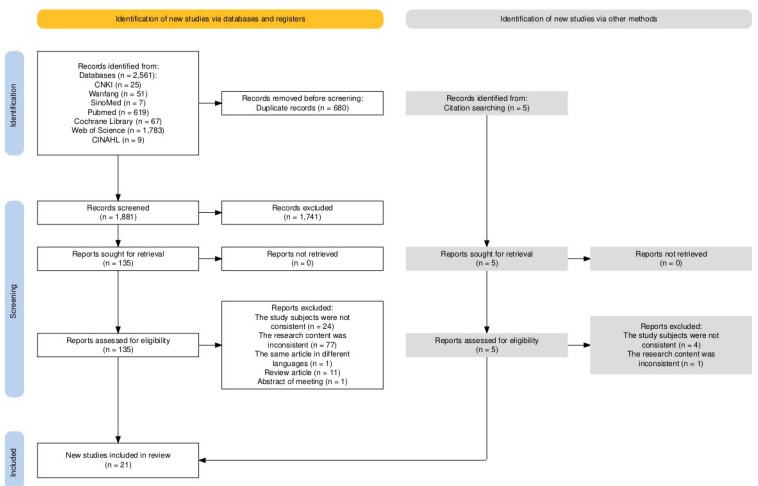

**Fig 1. Flow chart of literature selection.**

## Descriptive characteristics of the reviewed articles

Overall, 21 articles involving 20 transition readiness assessment tools were included in this study. The earliest assessment tools were developed in 2011[12], and the most articles were published in 2015 (n = 4) [22–24, 26] and 2021 (n = 4) [18, 31–33]. A representation of the number of articles published per year is shown in Fig 2. These tools were developed by scholars from different countries, including the US (n = 8), China (n = 2), France (n = 2), Canada (n = 2), and other countries (n = 7).

## Development procedures of transition readiness assessment tools

According to the principles of scale development [34], a set of scientific assessment tools needs to go through six steps: literature review, qualitative interviews, the Delphi method, group discussion, item analysis, and reliability and validity tests. Table 1 shows the development process of these 20 tools. Some assessment tools lacked some certain steps. Five assessment tools

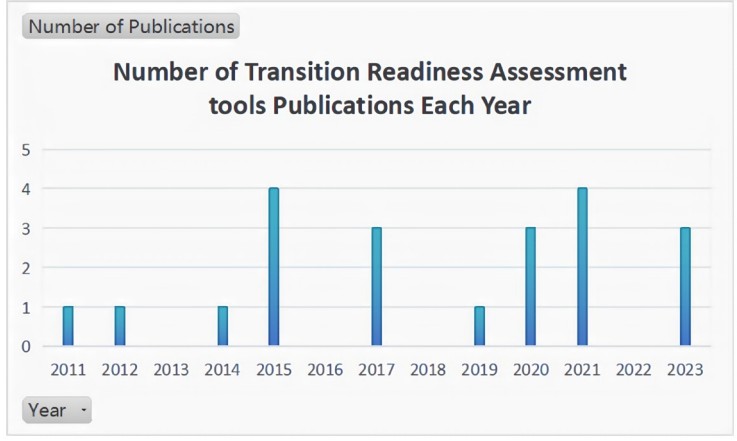

**Fig 2. Number of publications per year.**

**Table 1. Development procedures of the transition readiness assessment tools (n = 20).**

| | Name of tool | Year -author | Literature review | Qualitative interview | Delphi Method | Group discussion | Item analysis | Reliability and validity test |
|---|---|---|---|---|---|---|---|---|
| TRAQ | TRAQ-29items [12] | 2011 Sawicki | √ | × | √ | √ | √ | √ |
| | TRAQ-20items [14] | 2014 Wood | NS[a] | NS | NS | NS | √ | √ |
| | Argentinian Version [15, 16] | 2017De Cunto, 2017González | NS | NS | NS | NS | NS | √ |
| | Japanese version [17] | 2020Yuki Sato | NS | NS | NS | NS | √ | √ |
| | TRAQ20-Item New [18] | 2021Johnson | NS | NS | NS | NS | √ | √ |
| | Hungarian Version [19] | 2023Dóra | NS | NS | NS | NS | NS | √ |
| | TRAQ-NL [20] | 2023van Gaalen | NS | NS | NS | NS | NS | √ |
| UNCTRxANSITION Scale [21] | | 2012 Ferris | √ | √ | × | √ | √ | √ |
| ADAPT [22] | | 2015 Sawicki | √ | √ | √ | √ | × | √ |
| STARx | Initial Version [23, 24] | 2015Ferris, 2015Sarah | √ | √ | √ | √ | √ | √ |
| | Chinese Version [25] | 2023Yunzhen Huang | NS | NS | NS | NS | √ | √ |
| | Hungarian Version [19] | 2023Dóra | NS | NS | NS | NS | × | √ |
| Revised ON TRAC [26] | | 2015Melissa | NS | NS | NS | × | √ | √ |
| THRxEADS [27] | | 2017Nicholas | × | × | × | × | NS | NS |
| TRM [28] | | 2019 Hammerman | √ | × | √ | × | √ | √ |
| State assessment questionnaire for transition [29] | | 2020Funes D | √ | × | √ | √ | × | √ |
| Good2 Go [30] | | 2020Hélène | NS | NS | NS | NS | √ | √ |
| Checklist for follow-up of adolescents with chronic illness [31] | | 2021C Fourmaux | √ | × | × | √ | NS | NS |
| RAISE [32] | | 2021Susan | √ | √ | √ | √ | √ | √ |
| SASTRA [33] | | 2021Jing Huang | √ | √ | √ | √ | √ | √ |

[a], not suitable.

TRAQ, Transition Readiness Assessment Questionnaire; TRAQ-NL, Dutch Version of Transition Readiness Assessment Questionnaire;
ADAPT, Adolescent Assessment of Preparation for Transition; STARx, The Self-Management and Transition to Adulthood withRx = Treatment; TRM, "Transition Readiness Measure" for adolescents with IBD; Revised ON TRAC: Revised Am I ON TRAC for Adult Care Questionnaire; RAISE, Readiness Assessment of Independence for Specialty Encounters; SASTRA, Self-assessment Scale of Transition readiness for Adolescents.

(TRAQ-29items [12], THRxEADS [27], TRM [28], State Assessment Questionnaire for Transition [29], and Checklist for Follow-up of Adolescents with Chronic Illness [31]) did not include qualitative interviews. Three assessment tools (the UNCTRxANSITION Scale [21], THRxEADS [27], and Checklist for Follow-up of Adolescents with Chronic Illness [31]) lacked Delphi methods. Three tools (Revised ON TRAC [26], THRxEADS [27], and TRM [28]) did not include group discussion. Additionally, two tools (the STARx Hungarian Version [19] and the State Assessment Questionnaire for Transition [29]) skipped item analysis.

## Design of transition readiness assessment tools

Of the 20 transition readiness assessment tools, 2 tools [27, 31] were checklists, and the others were scales. There are some similarities and differences in the design of these tools. Most tools rely on patient self-reports, and only 4 tools [21, 27, 31, 32] use dual cross-referencing of patient statements with medical records. Most tools focused on medication management and

self-management, with some examining other aspects. Only one of the 20 tools was designed for IBD [28]. Specific information is given in Table 2.

## Psychometric properties of transition readiness assessment tools

The assessment of the methodological quality of the validation studies and the psychometric measurement qualities of the tools were integrated using Terwee's criteria checklist [35]. The checklist includes explicit criteria for the following measurement properties: content validity, internal consistency, criterion validity, construct validity, reproducibility, responsiveness, floor and ceiling effects, and interpretability. Criterion validity was removed from the analysis because there is no gold standard for measuring transition readiness, and all correlations were with theoretically derived hypotheses (construct validity). More attention was given to content validity, internal consistency, and construct validity when making a quality assessment. Most tools scored poorly according to the Terwee criteria, as shown in Table 3.

## Cohort characteristics for validity testing

The universality assessment tool had a diverse study population and was partially validated for those with IBD. Most of the tools have been validated in multicenter cross-sectional studies, with a few using a single center [15, 16, 20, 25, 26, 29, 33]. The age range of the validation population was large, ranging from 10 years old to 26 years old. The country of validation is mainly the USA. Table 4 shows the specific information of the cohort characteristics used for validity testing.

## Discussion

Over the years, varied research has been published in terms of transition readiness. Different authors have agreed on the assessment of transition readiness as a relevant tool for the health field. A reliable and valid transition readiness tool may dissipate some of the uncertainty around the transition process and allow for tailoring of programs to suit patients' transition demands [36]. However, existing tools have some limitations in assessing transition readiness in adolescents with IBD.

The scientific nature of the development process is critical to the assessment tool. Most of the assessment tools followed the six necessary steps for scientific accuracy. However, it was observed that some tools lacked certain key steps in their development, which may have some adverse effects. The absence of qualitative interviews may result in issues such as an inadequate construction of the scale's entry pool, an insufficient representation of its content, a lack of depth of data, a disconnection between theory and experience, and a limited scope of application [37, 38]. Consequently, in the process of scale development, researchers should prioritise the role of qualitative interviews and ensure the scientific and practicality of the scale through in-depth qualitative research. Three assessment tools lacked Delphi methods, which may result in a reduction in the comprehensiveness and accuracy of the entries, as well as an impairment of the scale's scientific and authoritative nature [39]. The Delphi methods is crucial for scale construction. It enhances the scientific rigour, credibility and practicality of the scale, thereby facilitating its wider application and dissemination [40]. Therefore, researchers should fully utilize the Delphi method throughout the scale construction process to guarantee the quality and practicality of the scale. And lacking group discussion may give rise to biases in the understanding of the subject matter and problems with the accuracy of data collection [41]. Additionally, it is of paramount importance to underscore the pivotal role of item analysis in the scale development process. Item analysis is an essential component of ensuring the reliability and effectiveness of a scale. It involves a comprehensive screening and optimisation process,

**Table 2. Design of transition readiness assessment tools (n = 20).**

| Name of tool | | Style | Disease | Language | Domain | Item (s) | Nature of responses | Reporter |
|---|---|---|---|---|---|---|---|---|
| TRAQ | TRAQ-29items [12] | Scale | Chronic diseases | English | 1.Self-management; 2.Self-Advocacy | 29 | 5-point Likert scale self-reported | Patient |
| | TRAQ-20items [14] | Scale | Chronic diseases | English | 1.Managing Medications; 2.Appointment Keeping; 3.Tracking Health Issues; 4.Talking With Providers; 5.Managing Daily Activities | 20 | 6-point Likert scale self-reported | Patient |
| | Argentinian Version [15, 16] | Scale | Chronic diseases | Spanish | 1.Managing Medication; 2.Appointment Keeping; 3.Tracking Health Issues; 4. Talking with Providers; 5.Managing Daily Activities | 20 | 5-point Likert scale self-reported | Patient |
| | Japanese version [17] | Scale | Chronic diseases | Japanese | 1.Managing medications; 2.keeping appointment; 3.tracking health issues; 4. Talking with providers | 23 | 6-point Likert scale self-reported | Patient |
| | TRAQ20-Item New [18] | Scale | Chronic diseases | English | 1.Managing medications; 2.Keeping appointment; 3.Tracking health issues; 4.Talking with providers | 20 | 5-point Likert scale self-reported | Patient |
| | Hungarian Version [19] | Scale | Chronic diseases | Hungarian | 1.Managing Medication; 2.Appointment Keeping; 3.Tracking Health Issues; 4. Talking with Providers; 5.Managing Daily Activities | 20 | 6-point Likert scale self-reported | Patient |
| | TRAQ-NL [20] | Scale | Chronic diseases | Dutch | 1.Managing medications; 2.Keeping appointment; 3.Tracking health issues; 4.Talking with providers | 20 | 5-point Likert scale self-reported | Patient |
| UNCTRxANSITION Scale [21] | | Scale | Chronic diseases | English | 1.Type of illness; 2.Rx = medications;3. Adherence; 4.Nutrition; 5.Self-management; 6.Informed-reproduction; 7.Trade/school; 8.Insurance; 9.Ongoing support; 10. New health providers. | 33 | Interview style cross-referenced with medical records | Patient |
| ADAPT [22] | | Scale | Chronic diseases | English | 1.Counseling on Transition Self-Management; 2. Counseling on Prescription Medication; 3.Transfer Planning | 26 | Dichotomous responses self-reported | Patient |
| STARx | Initial Version [23, 24] | Scale | Chronic diseases | English | 1.Medication management; 2.Provider communication; 3.Engagement during appointments; 4.Disease knowledge; 5. Adult health responsibilities; 6. Resource utilization | 18 | 6-point Likert scale self-reported | Patient |
| | Chinese Version [25] | Scale | Chronic diseases | Chinese | 1.Self-management; 2.Disease knowledge; 3.Provider communication | 13 | 5-point Likert scale self-reported | Patient |
| | Hungarian Version [19] | Scale | Chronic diseases | Dutch | 1.Medication management; 2.Provider communication; 3.Engagement during appointments; 4.Disease knowledge; 5. Adult health responsibilities; 6. Resource utilization | 18 | 6-point Likert scale self-reported | Patient&Parent |
| Revised ON TRAC [26] | | Scale | Chronic diseases | English | 1.Knowledge; 2.Behaviour | 25 | Domain1: 4-point Likert scale; Domain2: 5-point Likert scale self-reported | Patient |
| THRxEADS [27] | | Checklist | Chronic diseases | English | 1.T –Transition; 2.H –Home; 3.Rx–Medication and Treatment; 4.E –Education and Eating; 5.A –Activities and Affect; 6.D –Drugs; 7.S –Sexuality | 30 | Interview style cross-referenced with medical records | Health Care Provider |
| TRM [28] | | Scale | IBD | Hebrew | 1.Perceived knowledge regarding illness; 2.Perceived self-efficacy; 3.Perception of medical care | 16 | 5-point Likert scale self-reported | Patient&Parent&Health Care Provider |

*(Continued)*

**Table 2.** (Continued)

| Name of tool | Style | Disease | Language | Domain | Iteam (s) | Nature of responses | Reporter |
|---|---|---|---|---|---|---|---|
| State assessment questionnaire for transition [29] | Scale | Chronic diseases | Spanish | 1.Daily activities; 2.Aspects of my illness; 3.Management and use of medications; 4.Practical aspects of health care; 5. Involvement in the health checkup; 6. Transfer | 24 | 6-point Likert scale self-reported | Patient |
| Good2 Go [30] | Scale | Chronic diseases | French | 1.Health self-advocacy; 2.Knowledge about chronic conditions; 3.Self-management skills | 20 | 5-point Likert scale self-reported | Patient |
| Checklist for follow-up of adolescents with chronic illness [31] | Checklist | Chronic diseases | French | 1.HEADSS items; 2.Chronic illness items; 3.Clinical examination | 25 | Interview style cross-referenced with medical records | Medical Provider |
| RAISE [32] | Scale | Chronic diseases | English | 1.Disease Treatment and Healthy Living; 2. Disease Understanding and Communication | 123 | Interview style cross-referenced with medical records | Medical Provider |
| SASTRA [33] | Scale | Chronic diseases | Chinese | 1.Disease knowledge; 2.Medical review; 3.Medication management; 4.Health tracing; 5.Doctor/nurse–patient communication; 6.Self-management | 21 | 6-point Likert scale self-reported | Patient |

conducted using scientific methods [42]. The absence of item analysis can have a significant impact on the quality and practicality of the scale, making it essential to prioritise this during scale development. All in all, missing important steps may result in incomplete or unclear scale entries, which can undermine the validity of the scale.

The 20 assessment tools for transition readiness are multidimensional and comprehensive. Among the various dimensions, the most common were "medication management" (n = 13) and "self-management" (n = 7). This indicates that most developers of scales believe that adolescents with chronic illnesses should be knowledgeable about their medications and take responsibility for managing their illnesses as they transition to adult healthcare. As for adolescents with IBD, the administration of biological agents is a crucial aspect of maintaining the disease in remission [43]. Furthermore, adolescents in this state are more likely to achieve successful transition. Additionally, it is of significant importance to adolescents with IBD that they develop self-management skills as they transition into adulthood. It can not only assist in managing the disease, but also facilitate the development of mental health, social and professional competencies [44]. Nineteen of the 20 tools were designed for adolescents with chronic diseases in general, and their specific content lacked specificity for adolescents with inflammatory bowel disease. For adolescents with inflammatory bowel disease, the administration of biologics is a large tissue they cannot ignore; they need to be aware of their biologic type, frequency of dosing, and adverse effects, among other factors [45]. In addition, they should adjust their life routines, such as rest, diet and exercise, according to their disease status and maintain a good mindset. Except for the RAISE [32], the number of entries for the other assessment tools ranged from 13–33. This is a reasonable number and allows patients to complete the scales. The number of scale entries is a crucial factor in determining the quality and usefulness of the scale. It has been demonstrated that scales with an excess of entries can result in a reduction in the willingness of respondents to cooperate, a state of respondent fatigue, and an increase in the analytical complexity of the scale. Conversely, scales with an insufficient number of entries can lead to limitations in content validity and the emergence of unidimensional bias [46]. Most of the tools rely on patient self-assessment, with only 4 (UNCTRxANSITION

**Table 3. Summary of the assessment of the measurement properties of transition readiness tools by the Terwee criteria (n = 20).**

| Name of tool | | Content validity | Internal consistency | | Construct validity | Reproducibility | | Responsiveness | Floor or ceiling effect | Interpretability |
|---|---|---|---|---|---|---|---|---|---|---|
| | | | FA | Cronbach's alpha | | Agreement | Reliability | | | |
| TRAQ | TRAQ-29items [12] | + | + | +: total (0.93), domain 1 (0.92), domain 2 (0.82) | +: 100% (age, disease type, gender) | 0 | 0 | 0 | 0 | ?: no MIC defined |
| | TRAQ-20items [14] | 0 | + | -: domain 5 (0.67) | -: 50%(age, gender)no correlation with race or insurance | 0 | 0 | 0 | 0 | ?: no MIC defined |
| | Argentinian Version [15, 16] | 0 | 0 | -: domain 1 (0.60), domain 3 (0.30), domain 4 (0.52), domain 5 (0.55) | +: 80%(age, gender, perceived transition readiness, futuer plan) no correlation with health impairment due to the condition | 0 | 0 | 0 | + | ?: no MIC defined |
| | Japanese version [17] | + | 0 | +: total (0.94), 0.80 to 0.90 across the domains | +: 75%(age, knowledge of disease name, who accompanies hospital visits)no correlation with gender | 0 | 0 | 0 | - | ?: no MIC defined |
| | TRAQ20-Item New [18] | 0 | + | ?: Cronbach's alpha(s) calculated per dimension is not reported | 0 | 0 | 0 | 0 | 0 | 0 |
| | Hungarian Version [19] | 0 | - | -: domain 3 (0.546), domain 4 (0.577), domain 5 (0.622) | -: 50%(age, gender, treatments) no correlation with disease duration, ethnicity, disease type | - | 0 | 0 | - | ?: no MIC defined |
| | TRAQ-NL [20] | 0 | - | ?: Cronbach's alpha(s) calculated per dimension is not reported | +: 80%(age, gender, disease type, repeated TRAQ administration, VAS self-management, VAS transfer readiness, independency, accepted having IBD) no correlation with disease duration or educational level | ?: Methods is not clear. | 0 | 0 | 0 | ?: no MIC defined |
| UNCTRxANSITION Scale [21] | | + | 0 | ?: used PC | 0: inferred from development | ?: used age sensitivity | +:K = 0.71 (95% CI: 0.64, 0.77) | 0 | 0 | 0 |
| ADAPT [22] | | 0 | + | ?: used ordinal alpha(0.57–0.99) | +: Items are associated strongly with their hypothesized construct. | 0 | 0 | 0 | 0 | 0 |

(*Continued*)

**Table 3.** (Continued)

| Name of tool | | Content validity | Internal consistency | | Construct validity | Reproducibility | | Responsiveness | Floor or ceiling effect | Interpretability |
|---|---|---|---|---|---|---|---|---|---|---|
| | | | FA | Cronbach's alpha | | Agreement | Reliability | | | |
| STARx | Initial Version [23, 24] | + | + | -: domain 3 (0.62), domain 4 (0.69), domain 5 (0.55), domain 6 (0.44) | +: 100%(age) | ?:Small cohort (n = 26) | 0 | 0 | 0 | 0 |
| | Chinese Version [25] | + | + | +: total (0.83), 0.78 to 0.82 across the domains | -: 33.3%(age) no correlation with disease duration or gender | "+: ICC = 0.88, p < 0.001 | 0 | 0 | 0 | ?: no MIC defined |
| | Hungarian Version [19] | 0 | - | -: (STARx-P only domain3 (0.779)>0.7, STARx-A only domain3 (0.828), domain4 (0.720)>0.7 | -: 33.3%(disease duration, ethnicity) no correlation with age, gender, treatments, disease type | - | 0 | 0 | - | ?: no MIC defined |
| Revised ON TRAC [26] | | 0 | ?: only tested domain 1 14-item | ?: only tested domain 1 14-item | ?: no hypotheses, correlates with age and psychosocial maturity | 0 | 0 | 0 | 0 | ?: no MIC defined |
| THRxEADS [27] | | 0 | 0 | 0 | 0 | 0 | 0 | 0 | 0 | 0 |
| TRM [28] | | + | ?: Small cohort (n = 67) | +: total (0.93), 0.77 to 0.84 across the domains | ?: no hypotheses, age had a significant correlation with domain 2 scores. | 0 | 0 | 0 | 0 | 0 |
| State assessment questionnaire for transition [29] | | + | ?: only checked 9 of the 24 items | -: only domain3 (0.702)>0.7 | 0 | 0 | 0 | 0 | 0 | 0 |
| Good2 Go [30] | | 0 | + | +:0.72 to 0.85 across the domains | -: 33.3% (age) no correlation with gender or disease duration | +: test-retest reliability at 0.76, 0.70, and 0.80 for each domain | 0 | 0 | + | ?: no MIC defined |
| Checklist for follow-up of adolescents with chronic illness [31] | | 0 | 0 | 0 | 0 | 0 | 0 | 0 | 0 | 0 |
| RAISE [32] | | + | ?: used FVI | 0 | 0 | 0 | + | 0 | 0 | 0 |
| SASTRA [33] | | + | + | +: total (0.821), 0.806~0.868 across the domains | 0 | ?: Small cohort (n = 40) | 0 | 0 | 0 | 0 |

Rating: + = positive;? = intermediate; - = negative; 0 = no information available. FA, Factor Analysis; PC, Pearson's Correlations; K, Weighted-kappa; MIC, Minimal Important Change; ICC, Intra-class Coefficient; FVI, Factorial Validity Index.

Scale [21], THRxEADS [27], Checklist for Follow-up of Adolescents with Chronic Illness [31], and RAISE [32]) using dual cross-referencing of patient statements and medical records. Although self-reporting is an economical and simple method, its accuracy cannot be guaranteed. Therefore, it is recommended that an assessment tool be used that combines subjective evaluations with objective results.

**Table 4. Cohort characteristics for validity testing.**

| Name of tool | | Number | Age(years) | Disease type | Amount of setting | Race & gender | Country of validation |
|---|---|---|---|---|---|---|---|
| TRAQ | TRAQ-29items [12] | 192 | 16–26 (mean: 19.7) | Activity Limiting Physical Condition, Cognitive Impairment, Mental Health Condition | 3 | 64% white, 56% female | USA |
| | TRAQ-20items [14] | 526 | 14–21 (83.1%>18) | NRᵃ | 3 | 49.0% white, 39.6% African American | USA |
| | Argentinian Version [15, 16] | 191 | 14–26 (mean: 16.9) | G&D; BMT; Pulmo.; Onco.; MMC; DBT; Gastro.; Neuro.; Liver Tx | 1 | 52.4% female | Argentina |
| | Japanese version [17] | 76 | 16-20(mean:17.8 for male, 18.2 for female) | CKD; CHD and others. | 3 | 52.6% male | Japan |
| | TRAQ20-Item New [18] | 386 | 16-24(mean: 20) | NR | 3 | 87% White; 54% female | USA |
| | Hungarian Version [19] | 111 | 15-19(mean: 17) | IBD | 9 | 54% female | Hungary |
| | TRAQ-NL [20] | 136 | 16–18 | IBD | 1 | 58.1% male | Netherlands |
| UNCTRxANSITION Scale [21] | | 128 | 12-20(mean: 16.5) | IBD; CKD; HTN; Renal Tx; SLE; SCD; Diabetes mellitus | NR | 65% female | USA |
| ADAPT [22] | | 1648 | 15–18 | Complex chronic disease or Noncomplex chronic disease | 3 | 43.5% White, 28.5% Hispanic or Latino; 53.2% female | USA |
| STARx | Initial Version [23, 24] | 194 | 12-25((mean: 17.5) | CKD; IBD; End-stage KD; CF; SLE;SCD; HTN; HIV | NR | 56% Caucasian; 52% male | USA |
| | Chinese Version [25] | 624 | 10-24(mean: 19.66) | Respi.; Gastro.; Urinary; Endocrine; Neuro.; Circu.; Hematological diseases; Ear nose throat diseases; Eye diseases; Musculoskeletal diseases; Multisystem disease; Skin diseases | 1 | 55.6% male | China |
| | Hungarian Version [19] | 112 | 15-19(mean: 17) | IBD | 9 | 54.5% female | Hungary |
| Revised ON TRAC [26] | | 200 | 12-19(mean: 15.33) | Diabetes; CHD; Gastro.; Neuro. | 1 | 57% male | Canada |
| THRxEADS [27] | | - | - | - | - | - | - |
| TRM [28] | | 63 | 12-21(mean: 16.6) | IBD | 4 | 56% female | Israel |
| State assessment questionnaire for transition [29] | | 168 | 12-19(mean: 14.4) | Onco.; Pulmo.; Plastic Surgery;Endocrino.; Nephro.; Immuno.; Gastro. | 1 | 66% female | Chile |
| Good2 Go [30] | | 321 | 14-18(mean: 16.4) | Type 1 diabetes; Epilepsy; CF, JIA; IBD | 13 | 51% male | France and Canada |
| Checklist for follow-up of adolescents with chronic illness [31] | | - | - | - | - | - | - |
| RAISE [32] | | 36 ᵇ | NR | - | NR | 82.6% female | USA |
| SASTRA [33] | | 582 | 14-16(mean: 15.02) | Respi.; Digestive; Urinary; Endocrino.; Neuro.; Circu. | 1 | 52.9% female | China |

ᵃ, not reported;

ᵇ, the population is experts.

G&D, Growth & Development; BMT, Bone Marrow Transplantation; Pulmo., Pulmonology; Onco., Oncology; MMC, Myelomeningocele; DBT, Diabetes; Gastro., Gastroenterology; Neuro., Neurology; Liver Tx, Liver Transplantation; CKD, Chronic Kidney Disease; Renal Tx, Renal Transplant; CHD, Chronic Heart Disease; HTN, Hypertension; SLE, Systemic Lupus Erythematosus; SCD, Sickle Cell Disease; IBD, Inflammatory Bowel Disease; JIA, Juvenile Idiopathic Arthritis; HIV, Human Immunodeficiency Virus; Respi., Respiratory; Circu., Circulatory; Endocrino, Endocrinology; Nephro., Nephrology; Immuno., Immunology.

This scoping review shows that the psychometric properties of 20 available transition readiness tools are limited or untested. Only TRAQ-29 items [12] received positive ratings for the most important measurement properties: content validity, internal consistency, and construct validity. Some of the assessment tools [14–16, 19, 23, 24, 29] have aggregate scale Cronbach's α coefficients that exceed 0.7, while the subscale Cronbach's α coefficients fall below 0.7. When using these scales, it is important to concentrate on the overall score rather than individual parts [47]. Three of the measurement tools [12, 17, 28] were found to have internal consistency in the applicable population, and their Cronbach's α coefficients were higher than 0.90. This suggests that there may be item encumbrance in the tools, which presents ethical and practical problems related to answering burdens in large sample surveys [48]. Hence, further research is required to refine these tools and address the challenges posed by item encumbrance. The content validity was assessed by assigning values to the importance of the entries by experts in each field. The content validity of most tools was good, but some did not report it. If content validity is not reported, other researchers will not know if the scale covers all the concepts it's supposed to. This calls into question the integrity of the scale. A lack of transparency may lead other researchers to doubt and reject the scale thus affects the application and dissemination of the scale [49]. The construct validity of a questionnaire is the degree to which it measures what it is intended to measure based on theoretical assumptions. Most of the tools have been validated by measures of age, gender, disease type, and disease duration. As much of the value of a transition readiness tool is in its ability to time transition for optimal health outcomes, a longitudinal study of the tool's ability to predict future transition outcomes is necessary. These outcomes could include the number of hospital admissions, number of surgeries and so on [50]. Three assessment tools [25, 30, 33] showed good retest reliability (one [33] had a small sample), while the rest lacked evidence of consistency over time. According to the data analysis, the sensitivity to age is usually high, but the time stability is low. To make it easier to determine the transition target of different age groups, it is recommended to refine the age grouping in the data analysis. Only one assessment tool [21] showed good interrater reliability. Inter-rater reliability plays a crucial role in ensuring consistency and reliability of scoring results [51]. Floor or ceiling effects were ignored by most tools, with only five tools [15–17, 19, 30] reporting them, two [15, 16, 30] of which received positive ratings. These two effects are where the range of the response indicator is not large enough and the response stays at the top or bottom of the indicator scale, thus suffering a loss of validity of the indicator [52]. This means that the content analyzed by the scale may not effectively differentiate between individuals. In terms of interpretability, most tools compare the mean and standard deviation of patients' transition readiness in different groups, but none of these tools provide a definition of minimal important change (MIC). Moreover, none of these tools were studied in terms of decision values, and the threshold or its other form of reference score can also be used as an explanatory reference for future studies.

The validation studies originate from the USA, Canada, France, Hungary, China, Japan, Argentina, Israel, Chile and the Netherlands, and most used multi-center authentication. The validity of specific content or overall scores needs to be tested in culturally diverse areas and in different health care settings. One difference in health care provision between nations is the ability of pediatric clinicians to continue to care for young adults [53, 54]. For example, in China, the licensing and funding arrangements are such that children's hospitals do not admit patients older than 18 years [55]. This raises questions about the validity of these tools in a country with a different healthcare system and supports the need for ongoing validation trials. The initial subjects of the 19 assessment tools were not IBD patients but were developed in patients with general chronic diseases. Due to the heterogeneity of the patient population, reliability and validity tests for IBD patients are essential. Only the TRAQ Hungarian Version

[19], TRAQ-NL[30], and STARx Hungarian Version[19] were used for the validation of the IBD patient population. The reliability and validity of the tool should be further verified before clinical use.

The transition readiness assessment tools are ultimately intended to be used in clinical settings, and they can effectively measure the level of transition readiness of these children with chronic disease. When utilizing these assessment tools, a series of principles must be adhered to. Firstly, the timing of the assessment should be tailored to the accessibility of adult health care facilities in various countries and conducted as early as feasible. For example, according to the transition practice guidelines in the United States [56], it is advisable to initiate discussions regarding transition-related policies at age 12. Preparation for this transition should occur between ages 14 and 18, with the official transfer to adult healthcare taking place between ages 18 and 21. Therefore, in the United States, transition readiness levels should be assessed starting from age 14. Furthermore, it is recommended that evaluations be conducted by professionals with specialized knowledge. Additionally, it is the recommended approach to compare the patient's self-assessment results with medical records. This method ensures that the assessment outcomes are more objective and accurate.

The results of this study suggest that there are common elements in the assessment tools of transition readiness in adolescents with IBD with other chronic physical health conditions, such as emphasizing self-management ability and medical skills [57]. Disease-specific measurement tools have their own unique features, such as the assessment tools for childhood cancer survivors, which adds the assessment of cancer-induced emotional problems and cancer recurrence [58]; Similarly, the transition in adolescents with IBD is unique in that it should focus on knowledge of the medications and its impact on the future. The challenges and gaps identified in transition readiness for IBD are unique in chronic disease management.

## Strengths and limitations

This study is the first one to map transition readiness assessment tools for adolescents with IBD from the aspects of development procedures, design, psychometric properties, and cohort characteristics for validity testing. Existing tools have several limitations in assessing transition readiness in adolescents with IBD. It would be appropriate for future studies to exam the validility and reliabibility of these tools in adolescents with IBD via more population studies. Besides, new assessment tools with a complete development process should be tailored to the characteristics of adolescents with inflammatory bowel disease. They should be consistent with the national healthcare context and use a large amount of demographic data to validate their scientific validity and effectiveness.

The main limitation that can be found throughout this study is that the inclusion of articles limiting the language to English and Chinese may have resulted in the loss of some scales published in other languages.

## Conclusion

The transitional readiness of adolescents with IBD plays an important role in the quality of life of children later in life. Although there are several valid instruments for screening transition readiness, all of these instruments have unique characteristics with strengths and weaknesses. Overall, the TRM is currently the most suitable assessment tool; however, its methodological quality remains to be further validated, and the accuracy of the results is limited by the manner in which the self-assessment test was completed. The most appropriate assessment tool to be used is the one that best suits individual conditions, accompanied by a comprehensive assessment of the patient. Despite the significant progress made in the field, more population studies

## Supporting information

**S1 Table. Content, reliability and validity of transition readiness assessment tools (n = 20).**
(XLSX)

**S2 Table. All studies identified through our literature search (n = 2566).**
(XLSX)

**S3 Table. All data extracted from each study for this scoping review (n = 21).**
(XLSX)

**S1 File. Search strategy.**
(PDF)

**S1 Checklist. Preferred Reporting Items for Systematic reviews and Meta-Analyses extension for Scoping Reviews (PRISMA-ScR) checklist.**
(PDF)

## Acknowledgments

We would like to thank the hospital's technical faculty for their guidance on methodology.

## Author Contributions

**Conceptualization:** YaHui Zuo, Mei Li, JinJin Cao.

**Data curation:** YaHui Zuo, Mei Li, JinJin Cao, JiaNan Wang.

**Formal analysis:** YaHui Zuo, Mei Li, JinJin Cao, JiaNan Wang, WenQian Cai.

**Methodology:** YaHui Zuo, Mei Li, JinJin Cao.

**Visualization:** YaHui Zuo, Mei Li, JinJin Cao, Lu Zhang.

**Writing – original draft:** YaHui Zuo.

**Writing – review & editing:** YaHui Zuo, Mei Li, JinJin Cao, Meng Li.

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
