## [Decision Letter · Decision Letter 0]

12 Nov 2024

PONE-D-24-28612Assessment tools for Transition Readiness in adolescents with Inflammatory Bowel Disease: A scoping reviewPLOS ONE

Dear Dr. Li,

Thank you for submitting your manuscript to PLOS ONE. After careful consideration, we feel that it has merit but does not fully meet PLOS ONE’s publication criteria as it currently stands. Therefore, we invite you to submit a revised version of the manuscript that addresses the points raised during the review process.

We look forward to receiving your revised manuscript.

Kind regards,

Sara Hemati

Academic Editor

PLOS ONE

Journal Requirements:

2. We note that your Data Availability Statement is currently as follows: “All relevant data are within the manuscript and in Supporting Information files.”

Please confirm at this time whether or not your submission contains all raw data required to replicate the results of your study. Authors must share the “minimal data set” for their submission. PLOS defines the minimal data set to consist of the data required to replicate all study findings reported in the article, as well as related metadata and methods (https://journals.plos.org/plosone/s/data-availability#loc-minimal-data-set-definition). For example, authors should submit the following data: - The values behind the means, standard deviations and other measures reported; - The values used to build graphs; - The points extracted from images for analysis. Authors do not need to submit their entire data set if only a portion of the data was used in the reported study. If your submission does not contain these data, please either upload them as Supporting Information files or deposit them to a stable, public repository and provide us with the relevant URLs, DOIs, or accession numbers. For a list of recommended repositories, please see https://journals.plos.org/plosone/s/recommended-repositories. If there are ethical or legal restrictions on sharing a de-identified data set, please explain them in detail (e.g., data contain potentially sensitive information, data are owned by a third-party organization, etc.) and who has imposed them (e.g., an ethics committee). Please also provide contact information for a data access committee, ethics committee, or other institutional body to which data requests may be sent. If data are owned by a third party, please indicate how others may request data access.

4. As required by our policy on Data Availability, please ensure your manuscript or supplementary information includes the following: A numbered table of all studies identified in the literature search, including those that were excluded from the analyses. For every excluded study, the table should list the reason(s) for exclusion. If any of the included studies are unpublished, include a link (URL) to the primary source or detailed information about how the content can be accessed. A table of all data extracted from the primary research sources for the systematic review and/or meta-analysis. The table must include the following information for each study: Name of data extractors and date of data extraction Confirmation that the study was eligible to be included in the review. All data extracted from each study for the reported systematic review and/or meta-analysis that would be needed to replicate your analyses. If data or supporting information were obtained from another source (e.g. correspondence with the author of the original research article), please provide the source of data and dates on which the data/information were obtained by your research group. If applicable for your analysis, a table showing the completed risk of bias and quality/certainty assessments for each study or outcome. Please ensure this is provided for each domain or parameter assessed. For example, if you used the Cochrane risk-of-bias tool for randomized trials, provide answers to each of the signalling questions for each study. If you used GRADE to assess certainty of evidence, provide judgements about each of the quality of evidence factor. This should be provided for each outcome. An explanation of how missing data were handled. This information can be included in the main text, supplementary information, or relevant data repository. Please note that providing these underlying data is a requirement for publication in this journal, and if these data are not provided your manuscript might be rejected.

Additional Editor Comments:Reviewer 1:

Dear Authors,

Thank you for your thorough and insightful scoping review on "Assessment Tools for Transition Readiness in Adolescents with Inflammatory Bowel Disease." This is an important area of research, and your paper makes a significant contribution by identifying and evaluating a broad range of tools used in clinical practice.

Some Points are important:

1. Depth of Analysis

- While your review is comprehensive, I encourage you to delve deeper into the analysis of each tool's psychometric properties and practical applicability. A more detailed comparison of these aspects would provide readers with clearer insights into the strengths and limitations of each tool.

2. Discussion on Implementation

- It would be beneficial to expand the discussion on how these tools can be implemented in clinical settings. Offering practical guidance or examples of successful implementation in healthcare systems would enhance the utility of your review for clinicians and healthcare providers.

3. Addressing Potential Bias

- You mentioned the exclusion of studies where full texts were unavailable. It might be helpful to provide more details on how many studies were excluded for this reason and discuss any potential bias this may introduce. This transparency would strengthen the validity of your findings.

4. Comparative Perspective

- Consider adding a section that compares your findings with similar scoping reviews or systematic reviews in other chronic diseases. This could provide a useful context and show whether the challenges and gaps identified in transition readiness for IBD are unique or part of broader trends in chronic disease management.

5. Future Directions

- While you have identified significant gaps in the current tools, it would be helpful to suggest specific areas for future research. This could include recommendations for developing new tools or improving existing ones, particularly in areas where current tools are lacking.

your paper is well-structured and presents valuable information for both researchers and practitioners. With a few enhancements, it could have an even greater impact on the field. I appreciate the work you've done and look forward to seeing how this research progresses.

Reviewer 2:

Accept

Reviewers' comments:

Reviewer's Responses to Questions

**Comments to the Author**

1. Is the manuscript technically sound, and do the data support the conclusions?

Reviewer #1: Yes

Reviewer #2: Yes

2. Has the statistical analysis been performed appropriately and rigorously? 

Reviewer #1: Yes

Reviewer #2: Yes

3. Have the authors made all data underlying the findings in their manuscript fully available?

Reviewer #1: Yes

Reviewer #2: Yes

4. Is the manuscript presented in an intelligible fashion and written in standard English?

Reviewer #1: Yes

Reviewer #2: Yes

5. Review Comments to the Author

Reviewer #1: Dear Authors,

Thank you for your thorough and insightful scoping review on "Assessment Tools for Transition Readiness in Adolescents with Inflammatory Bowel Disease." This is an important area of research, and your paper makes a significant contribution by identifying and evaluating a broad range of tools used in clinical practice.

Some Points are important:

1. Depth of Analysis

- While your review is comprehensive, I encourage you to delve deeper into the analysis of each tool's psychometric properties and practical applicability. A more detailed comparison of these aspects would provide readers with clearer insights into the strengths and limitations of each tool.

2. Discussion on Implementation

- It would be beneficial to expand the discussion on how these tools can be implemented in clinical settings. Offering practical guidance or examples of successful implementation in healthcare systems would enhance the utility of your review for clinicians and healthcare providers.

3. Addressing Potential Bias

- You mentioned the exclusion of studies where full texts were unavailable. It might be helpful to provide more details on how many studies were excluded for this reason and discuss any potential bias this may introduce. This transparency would strengthen the validity of your findings.

4. Comparative Perspective

- Consider adding a section that compares your findings with similar scoping reviews or systematic reviews in other chronic diseases. This could provide a useful context and show whether the challenges and gaps identified in transition readiness for IBD are unique or part of broader trends in chronic disease management.

5. Future Directions

- While you have identified significant gaps in the current tools, it would be helpful to suggest specific areas for future research. This could include recommendations for developing new tools or improving existing ones, particularly in areas where current tools are lacking.

your paper is well-structured and presents valuable information for both researchers and practitioners. With a few enhancements, it could have an even greater impact on the field. I appreciate the work you've done and look forward to seeing how this research progresses.

Best regards,

Reviewer #2: I would deeply thank you for invitation me to review the manuscript with title" Assessment tools for Transition Readiness in adolescents with Inflammatory Bowel Disease: A scoping review". In general, your manuscript is acceptable and valuable and I have no comment.

6. PLOS authors have the option to publish the peer review history of their article (what does this mean?). If published, this will include your full peer review and any attached files.

Reviewer #1: **Yes: **Dr. Ramtin Naderian

Ramtinndn@gmail.com

Reviewer #2: No

---

## [Author Response · Author response to Decision Letter 0]

1 Dec 2024

Dear editor and reviewers,

We feel great thanks for your professional review work on our article. As you are concerned, there are several problems that need to be addressed. According to your nice suggestions, we will incorporate the recommended changes into the manuscript.

We sincerely thank the academic editor and all reviewers for your constructive feedback that we have used to improve the quality of our manuscript. The editor and reviewers’ comments are laid out below in italicized font and specific concerns have been numbered. Our response is given in normal font and changes/additions to the manuscript are given in the blue text. Here is a point-by-point response to the academic editor and reviewers’ comments and concerns.

Academic editor:

1.Please ensure that your manuscript meets PLOS ONE’s style requirement, including for those file naming.

Response: Thank you for your kind comment. After double-checking the manuscript with the PLOS ONE’s style requirement, we made the following changes: the formatting of figure 1 and figure 2 was corrected via PACE. The others meets PLOS ONE’s style requirement.

2.We note that your Data Availability Statement is currently as follows: “All relevant data are within the manuscript and in Supporting Information files.” Please confirm at this time whether or not your submission contains all raw data required to replicate the results of your study. 

Response: Thank you for your kind concern. We confirm that our submission contains all raw data required to replicate the results of our study.

3.When completing the data availability statement of the submission form, you indicated that you will make your data available on acceptance. We strongly recommend all authors decide on a data sharing plan before acceptance, as the process can be lengthy and hold up publication timelines. 

Response: Thank you for your suggestion. We had some understanding bias when choosing data availability. Actually, we are very happy to share our data with the public. All authors declare that they agree a data sharing plan before acceptance.

4.As required by our policy on Data Availability, please ensure your manuscript or supplementary information includes the following: A numbered table of all studies identified in the literature search, including those that were excluded from the analyses. For every excluded study, the table should list the reason(s) for exclusion. If any of the included studies are unpublished, include a link (URL) to the primary source or detailed information about how the content can be accessed. A table of all data extracted from the primary research sources for the systematic review and/or meta-analysis. The table must include the following information for each study: Name of data extractors and date of data extraction Confirmation that the study was eligible to be included in the review. All data extracted from each study for the reported systematic review and/or meta-analysis that would be needed to replicate your analyses. If data or supporting information were obtained from another source (e.g. correspondence with the author of the original research article), please provide the source of data and dates on which the data/information were obtained by your research group. If applicable for your analysis, a table showing the completed risk of bias and quality/certainty assessments for each study or outcome. 

Response: Thank you for your suggestion. To ensure the rigor and transparency of our results, we have included a numbered table of all studies identified through our literature search(S2 Table), as well as a table of all data extracted from each study for this scoping review(S3 Table), in the supplementary information section. Besides, accroding to the requirements of scoping review, a table showing the completed risk of bias and quality/certainty assessments for each study is not applicable. This study have reported the assessment results of each tools. 

5.Please review your reference list to ensure that it is complete and correct. If you have cited papers that have been retracted, please include the rationale for doing so in the manuscript text, or remove these references and replace them with relevant current references. Any changes to the reference list should be mentioned in the rebuttal letter that accompanies your revised manuscript. If you need to cite a retracted article, indicate the article’s retracted status in the References list and also include a citation and full reference for the retraction notice.

Response: Thank you for your kind comments. We have reviewed our reference list and adapted the reference format to the journal's requirements: capitalization was modified; removed the month date and kept only the year of publication. And there are no papers that have been retracted. In response to feedback from reviewers, three references were incorporated into the text, necessitating a subsequent reorganization of the existing references. The newly added references are enumerated below: 

56.White P, Schmidt A, Ilango S, Shorr J, Beck D, McManus M. Six core elements of health care transitionTM 3.0: an implementation guide (Got Transition, 2020). https://gottransition.org/6ce/?leaving-ImplGuide-full.

57.Schwartz LA, Daniel LC, Brumley LD, Barakat LP, Wesley KM, Tuchman LK. Measures of readiness to transition to adult health care for youth with chronic physical health conditions: a systematic review and recommendations for measurement testing and development. J Pediatr Psychol. 2014;39(6):588-601. 

58.Otth M, Denzler S, Koenig C, Koehler H, Scheinemann K. Transition from pediatric to adult follow-up care in childhood cancer survivors-a systematic review. J Cancer Surviv. 2021;15(1):151-162. 

Request for change of author's affiliation

Due to a change in employment status of the corresponding author, Jinjin Cao, we hereby request to update her affiliation from the Department of Gastroenterology, Children’s Hospital of Nanjing Medical University, Nanjing, Jiangsu Province, China, to the Department of Nursing, Nanjing BenQ Medical Center, The Affiliated BenQ Hospital of Nanjing Medical University, Nanjing, Jiangsu Province, China. We kindly seek your consent to this modification. We deeply appreciate your efforts in addressing this matter.

Reviewer #1: 

Thank you for your thorough and insightful scoping review on "Assessment Tools for Transition Readiness in Adolescents with Inflammatory Bowel Disease." This is an important area of research, and your paper makes a significant contribution by identifying and evaluating a broad range of tools used in clinical practice. Some Points are important. Your paper is well-structured and presents valuable information for both researchers and practitioners. With a few enhancements, it could have an even greater impact on the field. I appreciate the work you've done and look forward to seeing how this research progresses.

Response: Thank you very much for your strong support for our work, you have provided us with very valuable advice to improve the quality of this paper! We have used your comments to revise the manuscript and have attached a point-by-point response to your comments.

1. Depth of Analysis

While your review is comprehensive, I encourage you to delve deeper into the analysis of each tool's psychometric properties and practical applicability. A more detailed comparison of these aspects would provide readers with clearer insights into the strengths and limitations of each tool.

Response: We appreciate your valuable suggestion. We fully concur that the psychometric properties and practical applicability of assessment tools are of paramount importance. Considering the extensive nature of the content at the time of compilation, a comparison of the psychometric properties of these tools has been incorporated into S1 Table. Content, reliability and validity of transition readiness assessment tools (n=20) for the reader's ease of reference. Regarding the practical applicability of these assessment tools, we have clarified it in Table 4. These two tables provide specific details to facilitate a better understanding of the strengths and weaknesses of each tool for the readers.

2. Discussion on Implementation

It would be beneficial to expand the discussion on how these tools can be implemented in clinical settings. Offering practical guidance or examples of successful implementation in healthcare systems would enhance the utility of your review for clinicians and healthcare providers.

Response: We sincerely appreciate the valuable comments. It is imperative that assessment tools be used in a clinical setting, and therefore, guidance on their clinical use is necessary. A new section on page 34-35 line 316-329 has been included in the text, in which the potential applications of these tools in clinical settings are discussed. The section reads as follows: “The transition readiness assessment tools are ultimately intended to be used in clinical settings, and they can effectively measure the level of transition readiness of these children with chronic disease. When utilizing these assessment tools, a series of principles must be adhered to. Firstly, the timing of the assessment should be tailored to the accessibility of adult health care facilities in various countries and conducted as early as feasible. For example, according to the transition practice guidelines in the United States, it is advisable to initiate discussions regarding transition-related policies at age 12. Preparation for this transition should occur between ages 14 and 18, with the official transfer to adult healthcare taking place between ages 18 and 21. Therefore, in the United States, transition readiness levels should be assessed starting from age 14. Furthermore, it is recommended that evaluations be conducted by professionals with specialized knowledge. Additionally, it is the recommended approach to compare the patient's self-assessment results with medical records. This method ensures that the assessment outcomes are more objective and accurate.”

3. Addressing Potential Bias

You mentioned the exclusion of studies where full texts were unavailable. It might be helpful to provide more details on how many studies were excluded for this reason and discuss any potential bias this may introduce. This transparency would strengthen the validity of your findings.

Response: Thank you for your kind comment. On this point we have to make a clarification. In our pre-established exclusion criteria, we mentioned the exclusion of studies where full texts were unavailable. However, we were able to access the full text of virtually all of the content-related literature retrieved. Therefore, there is no literature that has been excluded because of this.

4. Comparative Perspective

Consider adding a section that compares your findings with similar scoping reviews or systematic reviews in other chronic diseases. This could provide a useful context and show whether the challenges and gaps identified in transition readiness for IBD are unique or part of broader trends in chronic disease management.

Response: Thank you for your kind suggestion. We added a section on page 35 line 330-340 that compares our findings with similar scoping reviews or systematic reviews in chronic physical health conditions and cancer. The section reads as follows: “The findings of this research indicate that there exist shared components in the assessment tools for assessing transition readiness among adolescents with IBD and those with other chronic physical health conditions. These shared components include an emphasis on self-management capabilities and medical competencies [57]. Disease-specific assessment tools possess distinct characteristics; for instance, the assessment tools for childhood cancer survivors incorporate an assessment of cancer-induced emotional issues and the recurrence of cancer [58]. Similarly, the transition process for adolescents with IBD is distinctive, as it necessitates a focus on medication knowledge and its ramifications for the future. The challenges and deficiencies identified in relation to transition readiness for IBD are indicative of broader trends observed in the management of chronic diseases.”

5. Future Directions

While you have identified significant gaps in the current tools, it would be helpful to suggest specific areas for future research. This could include recommendations for developing new tools or improving existing ones, particularly in areas where current tools are lacking. 

Response: Thank you for your valuable comment. We think this is an excellent suggestion. A section has been incorporated into the paper, located on page 35-36, at line 342-353, which provides recommendations for subsequent research endeavors. The contents of this section are as follows: “This research serves as the inaugural endeavor to map transition readiness assessment instruments for adolescents with inflammatory bowel disease (IBD), encompassing aspects such as development procedures, design, psychometric properties, and cohort characteristics for the purpose of validity testing. Current instruments possess several limitations in assessing transition readiness among adolescents with IBD. Therefore, it is advisable for future studies to examine the validity and reliability of these instruments in adolescents with IBD through more extensive population-based research. Additionally, novel assessment instruments, characterized by a comprehensive development process, should be tailored to the unique characteristics of adolescents with IBD. These instruments must align with the national healthcare framework and utilize extensive demographic data to establish their scientific validity and effectiveness.”

Reviewer #2: 

I would deeply thank you for invitating me to review the manuscript with title" Assessment tools for Transition Readiness in adolescents with Inflammatory Bowel Disease: A scoping review". In general, your manuscript is acceptable and valuable and I have no comment.

Response: We express our sincere gratitude for acknowledging our work, as it provides our team with considerable confidence to proceed with this project. We deeply appreciate the time and efforts you have devoted.

According to the academic editor and reviewers’ comments, we have made modifications to our manuscript and supplemented extra data to make our results convicing. We appreciate for academic editor and reviewers’ warm work earnestly, and hope the correction will meet with approval. Thank you again for your positive comments and valuable suggestions to improve the quality of our manuscript.

Yours sincerely,

Mei Li

30 November 2024

Children’s Hospital of Nanjing Medical University

---

## [Editor Report · Decision Letter 1]

22 Dec 2024

Assessment tools for Transition Readiness in adolescents with Inflammatory Bowel Disease: A scoping review

PONE-D-24-28612R1

Dear Dr. Li,

We’re pleased to inform you that your manuscript has been judged scientifically suitable for publication and will be formally accepted for publication once it meets all outstanding technical requirements.

Kind regards,

Sara Hemati

Academic Editor

PLOS ONE

Additional Editor Comments (optional):

Accept
---

## [Editor Report · Acceptance letter]

26 Dec 2024

PONE-D-24-28612R1 

PLOS ONE

Dear Dr. Li, 

I'm pleased to inform you that your manuscript has been deemed suitable for publication in PLOS ONE. Congratulations! Your manuscript is now being handed over to our production team.

Kind regards, 

on behalf of

Dr. Sara Hemati 

Academic Editor

PLOS ONE